# EFFICIENTLY QUANTIFYING INDIVIDUAL AGENT IMPORTANCE IN COOPERATIVE MARL

## ABSTRACT

Measuring the contribution of individual agents is challenging in cooperative multi-agent reinforcement learning (MARL). In cooperative MARL, team performance is typically inferred from a single shared global reward. Arguably, among the best current approaches to effectively measure individual agent contributions is to use Shapley values. However, calculating these values is expensive as the computational complexity grows exponentially with respect to the number of agents. In this paper, we adapt difference rewards into an efficient method for quantifying the contribution of individual agents, referred to as Agent Importance, offering a linear computational complexity relative to the number of agents. We show empirically that the computed values are strongly correlated with the true Shapley values, as well as the true underlying individual agent rewards, used as the ground truth in environments where these are available. We demonstrate how Agent Importance can be used to help study MARL systems by diagnosing algorithmic failures discovered in prior MARL benchmarking work. Our analysis illustrates Agent Importance as a valuable explainability component for future MARL benchmarks.

## 1 INTRODUCTION

In recent years, multi-agent reinforcement learning (MARL) has achieved significant progress, with agents being able to perform similar or better than human players and develop complex coordinated strategies in difficult games such as *Starcraft* (Samvelyan et al., 2019; Vinyals et al., 2019), *Hanabi* (Foerster et al., 2019; Bard et al., 2020; Hu and Foerster, 2021; Du et al., 2021) and *Diplomacy* (Bakhtin et al., 2022). Furthermore, MARL has also shown promising results in solving real-world problems such as resource allocation, management and sharing, network routing, and traffic signal controls (Vidhate and Kulkarni, 2017; Brittain and Wei, 2019; Nasir and Guo, 2019; Spatharis et al., 2019; Liu et al., 2020; Zhao et al., 2020; Pretorius et al., 2020; Gu et al., 2021). These real-world settings are naturally formulated as cooperative MARL systems, where agents need to coordinate to optimise the same global reward.

One of the critical challenges in cooperative MARL is multi-agent credit assignment (Chang et al., 2003). Since agents typically receive a global reward for their joint actions, this makes determining individual agent contributions challenging. This need for correct attribution becomes especially important as more autonomous systems are deployed in the real world. The inherent complexity of these MARL systems impedes our understanding of decision-making processes and the motivations behind actions, hindering progress in this field. Improved credit assignment could play a vital role in comprehending agent behaviour and system-level decision-making, aiding in accountability, trust, fairness, and facilitating the detection of potential issues such as coordination failures, or unethical behaviour.

Credit assignment can be considered from a core algorithmic perspective, where components of reinforcement learning (RL) algorithms, such as the value function, are adapted to better decouple the impact of the actions of individual agents. Methods such as VDN (Sunehag et al., 2017a), COMA (Foerster et al., 2018), and QMIX (Rashid et al., 2018a) fall into this domain. However, since these algorithms are trained end-to-end through the use of function approximators, explainability is difficult, i.e. it is challenging to correlate specific agent actions to reward outcomes over time. Furthermore,

since these notions of agent impact are part of the RL algorithms themselves, it is not easy to transfer these between different algorithms.

Accurate credit assignment within a team of agents can also be seen as a form of explainable AI (XAI). XAI consists of machine learning (ML) techniques that are used to provide insights into the workings of models (Arrieta et al., 2020). It has been used across various domains in ML, and more recently in single-agent RL[1] (Glanois et al., 2021) and multi-agent systems (Heuillet et al., 2022). Following from (Arrieta et al., 2020; Glanois et al., 2021), we use the notion of explainability to refer to any external post-hoc methodology that is used to gain insights into a trained model. These techniques have the notable advantage of being able to be used across algorithms, often irrespective of their design or formulation.

Efforts to enhance explainability in RL have resulted in the development of various techniques (Juozapaitis et al., 2019; Madumal et al., 2020; Puiutta and Veith, 2020; Glanois et al., 2021; Heuillet et al., 2021; Vouros, 2022; Dazeley et al., 2023). In contrast, MARL lacks dedicated explainability tools, with only a limited number of works addressing this topic (Kraus et al., 2019; Boggess et al., 2022; Heuillet et al., 2022). One notable approach involves leveraging the Shapley value (Shapley, 1953), a metric derived from game theory, and adapting it to MARL to quantify agent contributions to the global reward (Heuillet et al., 2022). Although Shapley values have shown promise in MARL explainability, calculating these values is expensive as the computational complexity grows exponentially with respect to the number of agents.

In this paper, we highlight the need for employing explainable tools to help quantify credit assignment in cooperative MARL systems. We show that an averaged calculation of the difference reward (Wolpert and Tumer, 2001) across evaluation episodes, can be used as an effective metric for measuring an agent's contribution, which we refer to as the *Agent Importance*. Unlike Shapley values, the Agent Importance has a linear computational complexity (w.r.t. the number of agents) making it more efficient to compute. Through empirical analysis, we demonstrate a strong correlation between the Agent Importance values and the true Shapley values, while also empirically validating the scalability and computational advantage of this approach.

To showcase the practical use of Agent Importance, we revisit a previous benchmark in cooperative MARL (Papoudakis et al., 2021) and follow the standardised evaluation guideline proposed by (Gorsane et al., 2022) to reproduce key results from this benchmark under a sound protocol. We then proceed by applying Agent Importance to specific scenarios of interest as highlighted by the authors of this benchmark. This includes investigating: (1) why Multi-Agent Advantage Actor-Critic (MAA2C) (Mnih et al., 2016a; Papoudakis et al., 2021) outperforms Multi-Agent Proximal Policy Optimisation (MAPPO) (Yu et al., 2022) in the Level-Based Foraging (LBF) environment [2] (Albrecht and Ramamoorthy, 2015; Albrecht and Stone, 2019; Christianos et al., 2020); and (2) why parameter sharing between agents leads to improved performance (3) analyse agents' behaviour in case of heterogeneous settings. Using agent importance, we uncover that for (1) MAA2C achieves a more equal contribution among agents when compared to MAPPO, i.e. agents have a more similar importance to the overall team and therefore have a higher degree of cooperation; and that for (2) architectures without parameter sharing exhibit a higher variance in agent importance, leading to credit assignment issues and lower performance compared to architectures with parameter sharing. The source code to reproduce our analysis and compute the agent importance, as well as our raw experiment data is publicly available [3].

## 2    RELATED WORK

**Explainability in RL** With the surging popularity of Deep RL, which relies on black-box deep neural networks, there has been an increase in literature that attempts to enable human understanding of complex, intelligent RL systems (Juozapaitis et al., 2019; Madumal et al., 2020; Puiutta and Veith, 2020; Glanois et al., 2021; Heuillet et al., 2021; Vouros, 2022; Dazeley et al., 2023). Additionally,

---

[1] In this paper, we use the term "RL" to exclusively refer to *single-agent* RL, as opposed to RL as a field of study, of which MARL is a subfield.

[2] A somewhat surprising result since MAPPO uses importance sampling for off-policy correction and is expected to perform at least as well as MAA2C as it incorporates a clipping function based on importance sampling allowing data retraining without divergent policies.

[3] Data is accessible at the following link.

frameworks like ShinRL (Kitamura and Yonetani, 2021) and environment suites like bsuite (Osband et al., 2019) offer comprehensive debugging tools including state and action space visualizations and reward distributions, and carefully crafted environments for behavioural analysis in RL.

**Explainability in MARL** In contrast to explainable RL, there has been a limited amount of work focusing on explainability in MARL (Kraus et al., 2019; Boggess et al., 2022; Heuillet et al., 2022). Specifically, we are interested in explainability in the context of cooperative MARL with a shared, global reward and the aim is to effectively quantify credit assignment.

The challenges associated with measuring credit assignment in MARL have motivated researchers to explore the use of the **Shapley value** (Shapley, 1953). Originating from game theory, the Shapley value addresses the issue of payoff distribution within a "grand coalition" (i.e. a cooperative game) and quantifies the contribution of each coalition member toward completing a task. Specifically, consider a cooperative game $\Gamma = (\mathcal{N}, v)$, where $\mathcal{N}$ is a set of all players and $v$ is the payoff function used to measure the "profits" earned by a given coalition (or subset) $\mathcal{C} \subseteq \mathcal{N} \setminus \{i\}$, such that the marginal contribution of player $i$ is given by $\phi_i(\mathcal{C}) = v(\mathcal{C} \cup \{i\}) - v(\mathcal{C})$. The Shapley value of each player $i$ can then be computed as:

$$S_i(\Gamma) = \sum_{\mathcal{C} \subseteq \mathcal{N} \setminus \{i\}} \frac{|\mathcal{C}|!(|\mathcal{N}| - |\mathcal{C}| - 1)!}{|\mathcal{N}|!} \cdot \phi_i(\mathcal{C}). \tag{1}$$

Calculating Shapley values in the context of MARL presents two specific challenges: (1) it requires computing $2^{n-1}$ possible coalitions of a potential $n(2^{n-1})$ coalitions (with $|\mathcal{N}| = n$) which is computationally prohibitive and (2) it strictly requires the use of a simulator where agents can be removed from the coalition and the payoff of the same states can be evaluated for each coalition.

Despite its limitations, the Shapley value is able to alleviate the issue of credit assignment and help towards understanding individual agent contributions in MARL. As a result, numerous efforts have been undertaken to incorporate it as a component of an algorithm (Wang et al., 2020; Yang et al., 2020a; Han et al., 2022; Wang et al., 2022). However, in this work, we focus on the Shapley value as an explainability metric. One such approach is introduced in (Heuillet et al., 2022), where the authors utilise a Monte Carlo approximation of the Shapley value to estimate the contribution of each agent in a system, which we refer to here as **MC-Shapley**. This approximate Shapley value is computed as:

$$\hat{S}_i^{MC}(\Gamma) = \frac{1}{M} \sum_{m=1}^{M} (r_{\mathcal{C}_m \cup \{i\}} - r_{\mathcal{C}_m}) \approx S_i(\Gamma), \tag{2}$$

where M is the number of samples (episodes), $C_m$ is a randomly sampled coalition out of all possible coalitions excluding agent $i$, and $r_{\mathcal{C}_m \cup \{i\}}$ and $r_{\mathcal{C}_m}$ are the episode returns obtained with and without agent $i$ included in the coalition.

In essence, Heuillet et al. (2022) attempts to address the second limitation of the Shapley value, which involves removing agents from the environment. They propose three strategies for proxies of agent removal while computing the return $r_{\mathcal{C}_m}$. The first hypothesis is to provide the agent $i$ with a no-op (no-operation) action, the second is to assign the agent $i$ with a random action, and the third is to replace the action of agent $i$ with a randomly selected agent's action from the current coalition $C_m$. The paper's findings indicate that using the no-op approach yields the most accurate approximation of the true Shapley value. A primary limitation of this work is the dependence on a significant number of sampled coalitions, with each sample corresponding to a single episode. This characteristic has a notable impact on training speed, especially if the proposed approach is employed as an online metric for detecting the evolution of agents' contributions during system training.

**Difference Rewards.** Of central relevance to this work is difference rewards (Wolpert and Tumer, 2001; Agogino and Tumer, 2004; 2008; Devlin et al., 2014) which presents a method for estimating credit assignment within a system. It can be written as $D_i(z) = G(z) - G(z_{-i})$ where $D_i(z)$ is the difference reward for agent $i$, $z$ is a state or state-action pair depending on the application, $G(z)$ is the performance of the global system and $G(z_{-i})$ is the performance of a theoretical system that omits agent $i$. Any action taken that increases the difference reward $D_i(z)$ also increases $G(z)$ but will have a higher impact on the (typically unknown or hypothetical) individual reward for each agent compared to the global reward. It is from this property that we may determine the relative impact of each agent in a system.

## 3 AGENT IMPORTANCE

We compute the Agent Importance as an average of difference rewards and use it as an efficient estimate of the Shapley value. To ensure accuracy in our estimation, we emphasize the importance of utilizing an adequate number of samples. This is reminiscent of the MC-Shapley approach which uses Monte Carlo approximation over entire episodes (Heuillet et al., 2022). However, in this work, we show that such an approach to estimation is not necessary and instead, we compute difference returns over samples collected *per step*, rather than per episode, without the need to resample coalitions. We simply compute the difference reward for each agent at each timestep during evaluation and aggregate over all evaluation timesteps. This approach greatly improves the sample efficiency in estimation during online evaluation. Concretely, the **Agent Importance** is given by

$$\hat{S}_i^{AI}(\Gamma) = \frac{1}{T} \sum_{t=1}^{T} r^t - r_{-i}^t, \tag{3}$$

where $T$ is the number of timesteps in a full evaluation interval, $r^t$ is the team reward (i.e. the reward of the grand coalition), at timestep $t$ and $r_{-i}^t$ is the team reward when agent $i$ performs a no-op action.

Applying Equation 3 poses a technical challenge as it requires comparing rewards between agents based on the same exact environment state at a given timestep. In MARL, most simulators are not easily resettable and/or stateless, which makes measuring one reward and undoing that step and then measuring a second reward difficult [4]. To overcome this limitation, we adopt a simple solution outlined in Algorithm 1, where we create a copy of the environment for each agent to be able to compute the Agent Importance.

---

**Algorithm 1** Per timestep difference reward contribution in Agent Importance

---

**Require:** $t$: evaluation timestep, $marginal\_contribution$: dictionary
  1: $env\_copies \leftarrow$ deepcopy$(env, len(agents))$          ▷ Create deepcopies of the environment.
  2: $r^t \leftarrow$ env.step$(selected\_actions)$
  3: **for** $i$ in range$(len(agents))$ **do**
  4:     $actions\_with\_no\_op \leftarrow$ disable_actions$(selected\_actions, i)$
  5:     $r_{-i}^t \leftarrow$ env_copies$[i]$.step$(actions\_with\_no\_op)$
  6:     add_to_dict$(marginal\_contribution, i, (r^t - r_{-i}^t))$
  7: **end for**

---

## 4 CASE STUDY: USING AGENT IMPORTANCE TO ANALYSE A PRIOR BENCHMARK

Our case study setup is based on the work of (Papoudakis et al., 2021), which made a comparative benchmark of cooperative MARL algorithms. The study conducts evaluations and comparisons of multiple categories of MARL algorithms, covering Q-Learning, and policy gradient (PG) methods, across two paradigms: independent learners (ILs), and centralised training with decentralised execution (CTDE). The findings of this study align with those of (Gorsane et al., 2022), concluding that current MARL algorithms are most performant on the popular Multi-Particle Environment (MPE) (Lowe et al., 2017) and Starcraft Multi-Agent Challenge (SMAC) (Samvelyan et al., 2019) environments–with most algorithms achieving comparable performance, in some cases seemingly to the point of overfitting. Consequently, our main analysis focuses on the remaining two environments from this benchmark: LBF, and RWARE.

**Environments.** The Multi-Robot Warehouse (RWARE) (Christianos et al., 2020; Papoudakis et al., 2021) is a multi-agent environment that is designed to represent a simplified setting where robots move goods around a warehouse. The environment requires agents (circles) to move requested shelves (colored squares) to the goal post (dark squares) and back to an empty square as illustrated at

---

[4]We however do note, that this could easily be achieved with simulators written using pure functions in JAX (Freeman et al., 2021; Lange, 2022; Bonnet et al., 2023).

the top of Figure 1. Tasks are partially observable with a very sparse reward signal as agents have a limited field of view and are rewarded only upon a successful delivery.

Level-Based Foraging (LBF) (Albrecht and Ramamoorthy, 2015; Albrecht and Stone, 2019; Christianos et al., 2020) is a mixed cooperative-competitive game with a focus on inter-agent coordination illustrated at the bottom of Figure 1. Agents are assigned different levels and navigate a grid world where the goal is to consume food by cooperating with other agents if required. Agents can only consume food if the combined level of the agents adjacent to a given item of food exceeds the level of the food item. Agents are awarded points equal to the level of the collected food divided by their level. LBF has a particularly high level of stochasticity since the spawning position and level assigned to each agent and food are all randomly reset at the start of each episode.

In the original benchmarking work by Papoudakis et al. (2021), the authors used the popular Starcraft Multi-Agent Challenge (SMAC) (Samvelyan et al., 2019) environment. In our case study, we instead use SMAClite (Michalski et al., 2023), an environment designed to replicate SMAC faithfully, in Python. An illustration of SMAClite is given in Figure 1. SMAClite has similar system dynamics to SMAC but does not rely on the StarCraft 2 video game engine as a backend. Due to this SMAClite requires significantly less RAM making it more suitable for utilising parallel processing. This also means it can be used in conjunction with Python methods like `copy` which makes contribution analysis methods like simpler to implement.

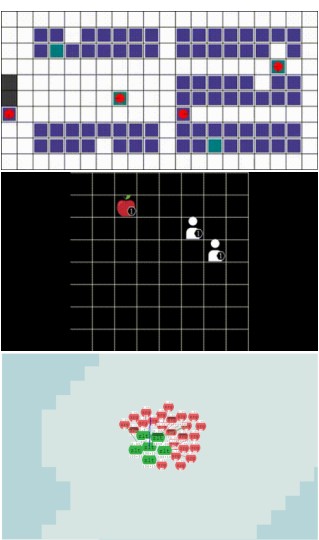

**Algorithms.** As in the original benchmarking setup of Papoudakis et al. (2021), we use the exact same collection of algorithms for our case study. Specifically, we use the value-based algorithms Independent Q-Learning (IQL) (Tan, 1997), Value-Decomposition Network (VDN) (Sunehag et al., 2017a), and QMIX (Rashid et al., 2018a), alongside two policy-gradient (PG) algorithms, namely Multi-Agent Proximal Policy Optimisation (MAPPO) (Yu et al., 2022) and Multi-Agent Advantage Actor-Critic (MAA2C) (Foerster et al., 2018). To investigate the influence of parameter sharing, we conduct experiments with both parameter-sharing and non-parameter-sharing architectures. Further details about the algorithms can be found in the Appendix section A.2.

Figure 1: **Top**: Multi-Robot Warehouse (RWARE). **Middle**: Level-Based Foraging (LBF). **Bottom**: SMAClite

**Evaluation Protocol.** We follow the protocol outlined by Gorsane et al. (2022), and apply the evaluation tools from Agarwal et al. (2022) in the MARL setting as advocated in the protocol. We evaluate agents at 201 equally spaced evaluation intervals for 32 episodes each during training. Following from the recommendations of Papoudakis et al. (2021) we train off-policy algorithms for a total of 2M timesteps and on-policy algorithms for a total of 20M timesteps summed across all parallel workers. This implies that evaluation occurs at fixed intervals of either 10k or 100k total environment steps for off- and on-policy algorithms respectively. For all our experiments, we use the EPyMARL framework (Papoudakis et al., 2021) which is opensourced under the Apache 2.0 licence. This is to ensure we are evaluating all algorithms on the same tasks, using the same codebase as was done by Papoudakis et al. (2021) for maximal reproducibility. Furthermore, it allows us to use identical hyperparameters as used in their work, which are available in the Appendix section A.3. All results that are presented are aggregated over 10 independent experiment trials. In cases where aggregations are done over multiple tasks within an environment, as opposed to an individual task (e.g. for computing performance profiles), the interquartile mean is reported along with 95% stratified bootstrap confidence intervals. For all plots except for sample efficiency curves, the absolute metric (Colas et al., 2018; Gorsane et al., 2022) for a given metric is computed. This metric is the average metric value of the best-performing policy found during training rolled out for 10 times the number of evaluation episodes.

**Computational resources**. All experiments were run on an internal cluster using either AMD EPYC 7452 or AMD EPYC 7742 CPUs. Each independent experiment run was assigned 5 CPUs and 5GB of RAM with the exception of the scalability experiments which were exclusively run using AMD EPYC 7742 CPUs and either 5, 15, 30, or 200 GB of memory depending on the number of agents and subsequently the number of environment copies that were required.

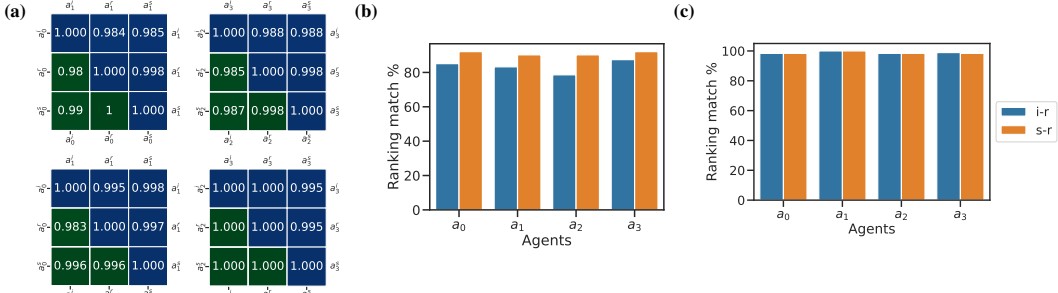

Figure 2: *Correlation analysis for agents* $\{a_0, a_1, a_2, a_3\}$, *for each metric: Agent Importance* $i$, *Shapley Value* $s$, *and Individual Reward* $r$ *using the VDN algorithm.* **(a)** Heatmap of Correlations among Metrics. **TOP:** LBF 15x15-4p-5f. **BOTTOM:** RWARE small-4ag. **(b)** Matching Rankings Comparison on LBF 15x15-4p-5f. **(c)** Matching Rankings Comparison on RWARE small-4ag. The legend refers to which metric is being compared to the individual agent rewards.

## 5 RESULTS

We demonstrate the validity of Agent Importance by considering its correlation to the true Shapley value, its computational scalability and its reliability in quantifying individual agent contributions. We then proceed to illustrate how Agent Importance may be used as an explainability tool.

### 5.1 VALIDATING AGENT IMPORTANCE

**Correlation between Agent Importance and the Shapley value.** We note that the Agent Importance metric is not mathematically equivalent to the Shapley value. It focuses on the grand coalition rather than all possible agent coalitions. However, through empirical study, we argue that Agent Importance is sufficient for capturing agents' contributions in the context of cooperative MARL.

To validate our assertion, we conduct experiments on both LBF and RWARE to empirically assess the correlation between Agent Importance and the Shapley value. We generate a heatmap that describes the correlation between the metrics for the VDN algorithm. Furthermore, we assess the ability of a metric to maintain the relative agent rankings according to each agent's individual rewards (which are not seen by the agents). If a metric gives the same ranking to agents, we count this as a positive result–implying that a higher-ranking match is better. While only results on VDN are displayed, the trend is consistent for all algorithms across various tasks. Further results to this end are given in the Appendix section D.

Figure 2 (a) shows that there exists a strong correlation between the Agent Importance, the Shapley value and the individual agent reward as calculated by the Pearson correlation coefficient. This indicates the effectiveness of both the Shapley value and Agent Importance in assessing agents' contributions, making them valuable substitutes for individual agent rewards in environments where such rewards are unavailable. Notably, Agent Importance showcases a promising ability to effectively replace both the Shapley value and individual rewards. While the Shapley value may provide greater consistency in ranking information when compared to the Agent Importance (as illustrated in Figures 2 (b,c)) where the frequency of ranking agreement between the individual reward and the contribution estimators is illustrated, it is important to note that Agent Importance is highly correlated with the individual reward and shows a minimal rate of non-matched rankings.

**Scalability of Agent Importance.** In order to validate the computational feasibility of the simplified Agent Importance against the full Shapley value we record the run time of both approaches on LBF tasks with $2, 4, 10, 20$, and $50$ agents. We run the algorithm without any training and compute the number of seconds it takes for agents to take a single environment step while computing each metric. The reported results here are the mean and standard deviation over 3 independent runs. The Shapley value became prohibitively slow as agent number was increased and required approximately 2 hours to measure a single step within the environment with 20 agents. Nonetheless, Figure 3 clearly illustrates how the Agent Importance is significantly more computationally efficient than the Shapley value.

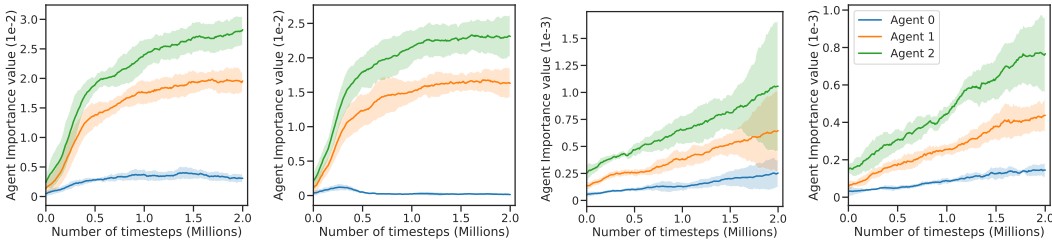

Figure 4: *Agent importance scores on the deterministic LBF scenario for MAA2C, MAPPO, VDN and QMIX.* Agents 0, 1 and 2 are assigned fixed levels of 1, 2 and 3 respectively–implying that their contributions should be weighted accordingly.

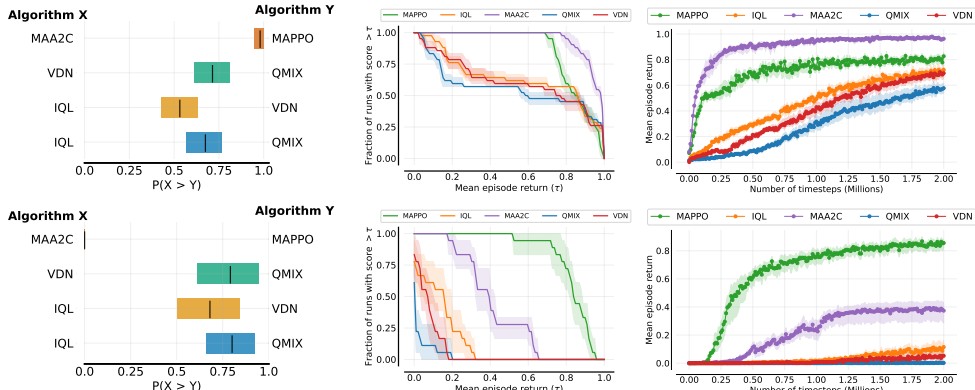

Figure 5: *Algorithm performance on LBF and RWARE including probability of improvement, performance profiles and sample efficiency curves.* **Top row:** Performance of algorithms on 7 LBF tasks. **Bottom row:** Performance of all algorithms on 3 RWARE tasks.

**Reliability of Agent Importance.** In order to validate the ability of the Agent Importance to effectively untangle agent contributions from a shared team reward, we create a deterministic version of LBF where agent levels are always fixed to be 1, 2, and 3 respectively, and the maximum level of each food is a random value between 1 and 6. Since agent 2 is assigned a fixed greater level than its counterparts we should expect it to contribute the most to the team return. Figure 4 illustrates the ability of Agent Importance to uncover the correct ordering and approximate level of contribution among agents towards the overall team goal.

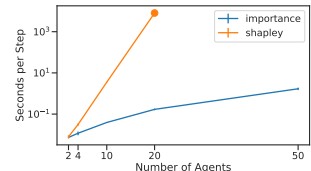

Figure 3: *Computational cost of computing the agent importance and the Shapley value.*

### 5.2 APPLICATIONS OF AGENT IMPORTANCE

We replicated the experiments performed by Papoudakis et al. (2021), obtaining similar results. However, our work adds value by following a strict protocol (Gorsane et al., 2022) which includes additional evaluation measurements such as examining the probability of improvement and providing performance profiles (Agarwal et al., 2022), as shown in Figure 5. Additional plots and tabular results for different scenarios and the performance of the algorithms without parameter sharing are included in the Appendix along with more detailed performance plots for SMAClite in Appendix section C.

**MAA2C vs MAPPO.** Empirical results in RL consistently demonstrate that PPO tends to outperform A2C (Heess et al., 2017; Schulman et al., 2017; Henderson et al., 2018). This trend naturally leads to the question of whether a similar pattern is observed in the multi-agent setting, i.e. between MAPPO and MAA2C. However, when examining the results in Figure 5, a conflicting observation arises. In the case of RWARE, we observe the expected behaviour with the probability of improvement

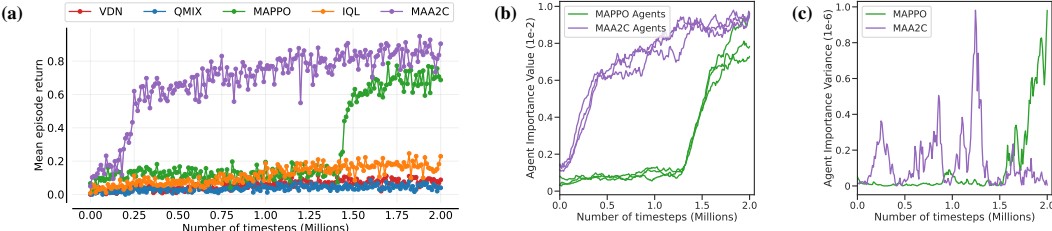

Figure 6: *MAA2C outperforms MAPPO on the LBF 15x15-3p-5f task.* **(a)** Sample efficiency curves (one seed) . **(b)** Agent importance for all agents associated with a given algorithm. **(c)** Variance of Agent Importance. MAA2C has lower variance in agent importance at convergence.

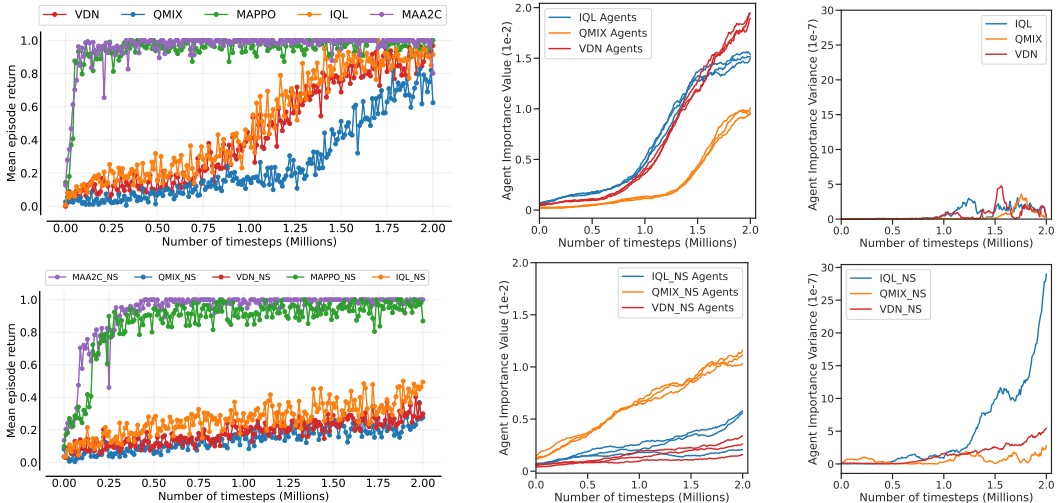

Figure 7: *Comparison of performance with and without parameter sharing on the LBF 10x10-3p-3f task for one seed including the sample efficiency, Agent Importance, and Agent Importance variance.* **Top row:** Performance with policy parameter sharing. **Bottom row:** Performance without policy parameter sharing. With parameter sharing the agent importance is more evenly distributed.

aligning with our initial expectations. However, in the case of LBF, the opposite occurs as MAA2C outperforms MAPPO, presenting an unexpected outcome. Figure 6 (b) highlights a possible reason. By tracking Agent Importance, we may attribute this outcome to a narrowing in the spread of importance values between MAA2C agents at convergence, as compared to MAPPO agents. The assumption of lower variance in Agent Importance leading to improved performance in LBF is due to the stochasticity of the environment. It is reasonable to expect that an algorithm performing well in this environment should have the capability to adapt to the variability in agent and food levels across episodes. From the narrower spread in Agent Importance values in MAA2C we can see it has learnt treat all agents as equally important.For additional findings on RWARE see section B.3 in the supplementary material.

**Parameter sharing vs non-parameter sharing.** Consistent with the findings of Papoudakis et al. (2021), our experiments demonstrate that algorithms utilizing parameter sharing outperform those without it. As mentioned in the benchmark paper, this outcome is expected as parameter sharing enhances sample efficiency. Additionally, parameter sharing enhances the sharing of learned information across the system. The Agent Importance analysis for IQL, QMIX, and VDN provides clear evidence of the impact of parameter-sharing architectures, as illustrated in Figure 7. It is apparent that in the absence of parameter sharing, the agents contribute to varying degrees, leading to an uneven distribution of importance. And as mentioned previously, given LBF's characteristics, requiring a high level of coordination in the presence of significant stochasticity, all agents should be expected (on average) to contribute equally. However, in the non-parameter sharing cases, especially for IQL and VDN, we observe that a small number of the agents dominate the contributions, resulting in lower performance compared to when parameter sharing is utilised.

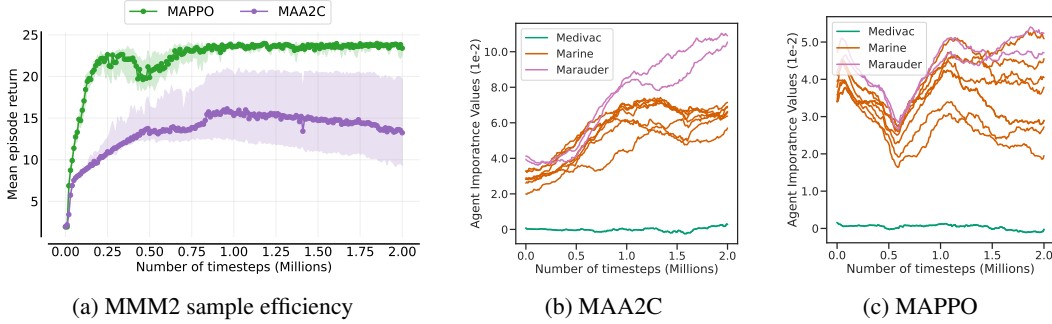

(a) MMM2 sample efficiency  (b) MAA2C  (c) MAPPO

Figure 8: *Comparison between MAA2C and MAPPO in the MMM2 scenario from SMAClite.*8a: Mean episode returns. 8b: Agent Importance scores MAA2C. 8c: Agent Importance scores MAPPO.

**Heterogeneous Agents.** In both LBF and RWARE the importance of each agent and the total reward are highly correlated as all agents have similar capabilities. In the heterogeneous setting of MMM2, rather than converging to similar importance levels over time, agents will instead converge to clear groups of importance levels as seen in figures 8b and 8c. Furthermore, note that agents of the same type can still fall into different levels of importance which is consistent with role decomposition analysis in ROMA (Yang et al., 2020b). As shown by Yang et al. (2020b), the optimal policy in MMM2 requires a subset of marine agents to die early in the episode, who then cannot contribute to the team reward remaining timesteps, whereas a smaller number of marines survive until the end. This is clearly seen in figure 8c for MAPPO. In the case of MAA2C in figure 8b we can see that although clear clusters have formed, it has not learned to assign the correct importance to a subgroup of marines that are required to optimally solve the environment [5] [6].

## 6 DISCUSSION

In this work, we illustrate that Agent Importance is an efficient and reliable measure for agent contributions towards the team reward in cooperative MARL. Aside from only quantifying the agent contributions we have also shown how the metric may be used as an explainability tool for uncovering failure modes in existing MARL results.

**Limitations.** Although Agent Importance is useful, using simulators that allow for an agent's removal during runtime would be highly advantageous. Solely relying on no-op actions could still impact the coalition reward by obstructing other agents' presence and movement in their observations. Unfortunately, agent removal is uncommon in most simulators and some simulators also do not offer the option for a no-op action. Additionally, while popular MARL research environments are fairly low resource, creating multiple parallel instances of the environment during the Agent Importance calculation, makes using more resource-heavy simulators prohibitive from a memory perspective. However, with the growing popularity of the JAX framework, more stateless environments are becoming available where the parallel environments can be replaced with direct access to the environment state (Freeman et al., 2021; Lange, 2022; Bonnet et al., 2023).

**Future Work.** It would be useful to investigate the rankings calculated by *agent importance* for simulators which do not have a no-op action. We could consider using random actions or the random actions of specific agents as a proxy for the no-op action or make use of function approximators to learn minimal impact actions for the marginalised agents.

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
