# OpenReview forum: "Efficiently Quantifying Individual Agent Importance in Cooperative MARL"
_ICLR.cc/2024/Conference — ICLR 2024 Conference Withdrawn Submission_

### Official Review · Reviewer_Vw4r · 2023-10-30

**Soundness:** 3 good
**Presentation:** 3 good
**Contribution:** 2 fair
**Rating:** 5
**Confidence:** 3

**Summary:**

The work addresses the explainability of MARL by proposing to use of difference rewards metrics instead of more complex metrics to be computed such as Shapley values and their Monte-Carlo sampled version. The work then proceeds to validate empirically the proposed metric with experiments on two environments.

**Strengths:**

The paper is well written and clear in the exposition, the quality of the experimental setting is high and I believe it offers a precious contribution to how MARL experimentation should be performed in deep settings.

**Weaknesses:**

Unfortunately, I have some doubts about the originality and the significance of the work per se. More specifically:
- I am not sure about the novelty of the proposed method (Agent Importance) in the MARL setting, and treatment of this theme would be required in the related works section. Without this, it is not clear to me what the contribution is compared to other different reward-based methods in MARL.
- The Agent Importance seems to me a sort of first-order approximation (in the time and agents' space) of the Shapley value. Due to this, I would be surprised if some correlation would not arise. Due to this type of approximation, I am worried that the empirical evidence provided by the experiments is something like "in these environments, the first order approximation is sufficient to reproduce the time/agent intercorrelations", further limiting the contribution of the paper.

**Questions:**

I would invite the authors to answer the two points raised in the weaknesses section. For the second point, in particular, I believe that the Markov Games instances provided by the environments are not able to test the main feature of the Agent Importance, namely what happens when either:
- specific coalitions out of the trivial ones (with or without a single agent) are able to accumulate more rewards (that's why the Shapley Value tests against all of them)
- actions of agents have a long-term effect on the joint performance (that's why time-step rewards are usually accumulated in the returns)

I would ask the authors to comment on this and evaluate whether it is possible to test the proposed method on some simple Markov Games with designed transitions so as to reproduce the above-mentioned cases or on some other environments.

---

### Official Review · Reviewer_D2kb · 2023-10-31

**Soundness:** 2 fair
**Presentation:** 2 fair
**Contribution:** 2 fair
**Rating:** 5
**Confidence:** 5

**Summary:**

This paper aims at interpreting multi-agent reinforcement learning (MARL) algorithms through identifying credit assignment to each agent. The main contribution of this work is improving the computational complexity of Shapley value that was proposed before to evaluate each agent's credit. Furthermore, this paper utilizes the Shapley value to explain the performance of several multi-agent reinforcement learning algorithms. In details, the final performance of MARL algorithms is shown to be matched with the pattern of credit assignment produced by Shapley value.

**Strengths:**

Originality
---
This paper proposes a method called Agent Importance to just consider the grand coalition and an interval of an episode, rather than the all possible coalitions and the whole episode in the previous work on using Shapley value for long-term decision making process.

Quality
---
The experimental parts are extensive and complete, spreading from verifying the positive correlation between Shapley value and the proposed Agent Importance, the scalability and reliability to its evaluation performance on MARL algorithms.

Clarity
---
This paper is generally well written, especially with details in experiments and a good motivation in Introduction.

Significance
---
This topic investigated in this paper is significant, since the interpretability is a critical part of trustworthy AI which is a common goal around world at present. However, due to the requirement of access to a twin environment, the application of the proposed method could be restricted.

**Weaknesses:**

1. This paper neglects important works (though mentioned in related works) [1,2] about extending Shapley value to MARL to explain and understand MARL algorithms. [1,2] did not only incorporate Shapley value into MARL algorithms, but also formulating a complete theoretical framework to validate the incorporation. Furthermore, the Monte Carlo method to approximate in MARL was firstly attempted in [1] and the similar expression in Eq. 2 was also mentioned in [1,2]. Then, I suggest that the authors check these two works carefully and treat them fairly.

2. Another weakness of this work is that it lacks an insight into the simplification of Shapley value proposed in this paper called Agent Importance. For example, the authors are encouraged to give some reasons (not just verifying its correlations empirically) to validate the simplification of Shapley value. I suggest the authors can check Chapter for Shapley value in [3] to acquire more information and insights, the simplified expression form proposed in this paper is in analogy of one form during the derivation of Shapley value.

Reference
---
[1] Wang, J., Zhang, Y., Kim, T. K., & Gu, Y. (2020, April). Shapley Q-value: A local reward approach to solve global reward games. In Proceedings of the AAAI Conference on Artificial Intelligence (Vol. 34, No. 05, pp. 7285-7292).

[2] Wang, J., Zhang, Y., Gu, Y., & Kim, T. K. (2022). Shaq: Incorporating shapley value theory into multi-agent q-learning. Advances in Neural Information Processing Systems, 35, 5941-5954.

[3] Chalkiadakis, G., Elkind, E., & Wooldridge, M. (2022). Computational aspects of cooperative game theory. Springer Nature.

**Questions:**

1. Addressing the weaknesses delivered above.
2. Since Agent Importance needs to access a twin environment, this would impede its applicability to broader areas that are lacking of exact simulators. Could the authors give some discussion about this issue?

---

### Official Review · Reviewer_18rV · 2023-11-01

**Soundness:** 2 fair
**Presentation:** 4 excellent
**Contribution:** 2 fair
**Rating:** 3
**Confidence:** 4

**Summary:**

The authors propose an adaptation of difference rewards, "agent importance", for efficient computation of individual agent contribution. Shapley values are generally treated as one of the best approaches for measuring agent contribution, but it is often prohibitively expensive. It is exponential in the number of agents because it involves considering every possible coalition (2^N). On the other hand, agent importance is linear in the number of agents because it simply checks the counterfactual with agent i removed against reality. The authors draw a comparison to MC-Shapley which performs Monte Carlo estimates over episodes, whereas agent importance is calculated per step. The authors' method crucially requires either access to a resettable simulator or performs deepcopies of the environment to compute counterfactuals without resetting the previous action. The method is evaluated in a variety of MARL benchmarks: RWARE, LBF, SMAClite. The experiments demonstrate strong Pearson correlation between agent importance and {Shapley values, individual reward}. A case study is performed, explaining that MAA2C counterintuitively outperforms MAPPO possibly because agent importance is more uniform in the former. A similar case study is done to explain that parameter sharing is useful possibly because agent importance is more uniform with shared parameters. The authors acknowledge limitations such as reliance on no-op actions or ability to simulate with an agent removed and computation required to maintain deepcopies when the simulator is not resettable.

**Strengths:**

The paper's methods are not particularly novel but the paper provides novel analysis and hypotheses of MAA2C vs MAPPO as well as parameter sharing vs not. The paper provides fairly thorough empirical analysis of the environments given, with coverage of both gridworlds/discrete environments and more complicated ones (SMAClite), and follows evaluation protocol established in the literature. There is a good analysis and comparison to related work and a good assessment of the relative strengths and weaknesses of each method. The paper is very clearly written and it was very easy for me to follow without much effort. There is documentation of compute used. There is also acknowledgement of some limitations of the method.

**Weaknesses:**

The idea is not particularly novel or deep; agent importance is simply computing difference reward averaged across an episode, which is what the standard computation does (minus averaging). It seems the novelty of the work lies in 1) demonstrating strong Pearson correlation between agent importance and {Shapley values, individual reward} and 2) performing case studies in a couple of environments to hypothesize that good performance is due to uniform agent importance.

The fundamental weakness is that agent importance only considers the grand coalition rather than all coalitions. The authors say empirically this isn't a concern for cooperative MARL but they only provide experiments on a small number of environments. If that claim is to be made, there needs to be justification provided for why these experiments are representative for cooperative MARL overall. I am not convinced as written.

As the authors acknowledge, they require either a no-op action or simulator callable with a subset of agents. This is not applicable to all environments.

The authors also acknowledge the added computation required by maintaining deepcopies of the environment for every agent at every timestep.

I am not strongly convinced that the reason MAA2C does well is because of more uniform agent importance. Why is more uniform agent importance necessarily good? It's possible that a task that requires only 1 agent to do in 10 steps is actually done with 5 agents each taking 5 steps. The agent importance of the former would be {R, 0, 0, 0, 0} and {R, R, R, R, R}, however the former is more efficient in terms of agent * time units.

**Questions:**

- can you describe the policies you observe the agents doing in each environment as well as provide the computed values?
- I expect individual agent reward to be a poor measure of contribution for an agent that plays an assistive role rather than generating reward themself. is there such behavior observed in any of the environments evaluated on, and if not, could you add one or more such experiments? I expect Shapley to do well in this setting, so I'm a bit surprised there is strong correlation between agent importance and BOTH shapley and individual reward.
- I did not understand the format of figure 2a. can you please explain (and maybe also make it bigger so readers of a printed copy can see the small text more clearly)?
- can you provide some text hypothesizing what kinds of environments and coordination schemes you think agent importance will do well in and poorly in?
- Shapley is exponential in # agents yet the graph is linear? is it because the exponential blowup occurs after it becomes infeasible to compute? can you provide a cutoff time? I'd expect to see an exponential trend even with {2, 4, 10, 20} agents.

---

### Official Review · Reviewer_HLQg · 2023-11-06

**Soundness:** 3 good
**Presentation:** 3 good
**Contribution:** 2 fair
**Rating:** 5
**Confidence:** 4

**Summary:**

This article provides a heuristic that estimates the contribution of an agent in a fully cooperative (joint reward) multi-agent system towards the collective goal. The authors provide a metric called Agent Importance (AI) that is inspired by the Shapley value from cooperative game theory, but that doesn't require the computation of differences across all possible episode compositions. Instead, the authors suggest that having just episodes with the grand coalition, and comparing against invalidating the focal agent is enough to get a reliable estimate of the Shapley. The authors validate their metric in several benchmarks, and show high correspondence between AI and Shapley, as well as other, learned, systems to assign agent contributions like Q-MIX or VDNs.

**Strengths:**

The article is clearly written and presented, despite some issues (see below). The proposed heuristic approximation of agent contributions is an important problem in fully cooperative multi-agent research. The vast majority of the effort has been given to learned solutions that produce a contribution estimation alongside training. However, there is a very important niche for which post-hoc metrics are valid: explainability. The authors motivate well the use of their metric as an explainability metric.

**Weaknesses:**

While using a post-hoc metric like Shapley or the proposed AI is beneficial in some circumstances, it should be mentioned that doing so requires extra (test time) compute. Algorithms like Q-MIX or VDN do not suffer from this issue, in the sense that they provide an estimation of contribution at any point in training.

The authors show a reasonable heuristic for computing agent contribution, however, this is perhaps too simple. I would have liked to see something a bit more principled. Maybe comparing the full Shapley with other sub-exponential systems to estimate variables. The presented AI seems to be the obvious simplest heuristic. It is novel, to my understanding, so it might be worth publishing nonetheless.

Some of the figure captions are unclear. Many have a left and right panel division that is not discussed.

The discussion of which benchmarks are used in which contexts is complex. The authors would benefit from clearing up which benchmarks are used when, or, better yet, just stick to some for _all_ the experiments.

The authors require a no-op action to be admitted by the environment in order to estimate the contribution of a agent to the global objective. While this might be OK in many scenarios, this is not always the case (e.g. in extensive form games). The authors briefly discuss how producing random actions might be an alternative, but dismiss it as having too much variance. Inclusion of these results would increase the transparency of the contribution.

For all of the discussion about benchmarks, the authors stick to only a small subset of traditional multi-agent cooperative benchmarks.

**Questions:**

Why is Figure 2 showing correlations between pairs of agents? I thought these were 4 player games, and $a_0$ to $a_4$ were agents in the game.

The caption of figure 5 is unclear. Probability of improvement is not defined.

Figures 6 and 7 don't explain the left and right portions of the plots.